# Evaluation and Bias Correction of the Secondary Inorganic Aerosol Modeling over North China Plain in Autumn and Winter

**Qian Wu** [1,2], **Xiao Tang** [1,3,*], **Lei Kong** [1,2], **Xu Dao** [4], **Miaomiao Lu** [5,6], **Zirui Liu** [1], **Wei Wang** [4], **Qian Wang** [7], **Duohong Chen** [8], **Lin Wu** [1], **Xiaole Pan** [1], **Jie Li** [1,3], **Jiang Zhu** [1,2] and **Zifa Wang** [1,2,3]

1  LAPC&ICCES, Institute of Atmospheric Physics, Chinese Academy of Sciences, Beijing 100029, China; wuqian@mail.iap.ac.cn (Q.W.); konglei@mail.iap.ac.cn (L.K.); liuzirui@mail.iap.ac.cn (Z.L.); wlin@mail.iap.ac.cn (L.W.); panxiaole@mail.iap.ac.cn (X.P.); lijie8074@mail.iap.ac.cn (J.L.); jzhu@mail.iap.ac.cn (J.Z.); zifawang@mail.iap.ac.cn (Z.W.)
2  University of Chinese Academy of Sciences, Beijing 100049, China
3  Center for Excellence in Regional Atmospheric Environment, Institute of Urban Environment, Chinese Academy of Sciences, Xiamen 361021, China
4  China's National Environmental Monitoring Centre, Beijing 100012, China; daoxu@cnemc.cn (X.D.); wangwei2015@cnemc.cn (W.W.)
5  State Environmental Protection Key Laboratory of Urban Ambient Air Particulate Matter Pollution Prevention and Control, College of Environmental Science and Engineering, Nankai University, Tianjin 300350, China; lumiaomiao1027@126.com
6  CMA-NKU, Cooperative Laboratory for Atmospheric Environment-Health Research, Tianjin 300074, China
7  Shanghai Environmental Monitoring Centre, Shanghai 200235, China; wangqian@sheemc.cn
8  State Environmental Protection Key Laboratory of Regional Air Quality Monitoring, Guangdong Environmental Monitoring Centre, Guangzhou 510308, China; chenduohong@139.com
*  Correspondence: tangxiao@mail.iap.ac.cn

**Abstract:** Secondary inorganic aerosol (SIA) is the key driving factor of fine-particle explosive growth (FPEG) events, which are frequently observed in North China Plain. However, the SIA simulations remain highly uncertain over East Asia. To further investigate this issue, SIA modeling over North China Plain with the 15 km resolution Nested Air Quality Prediction Model System (NAQPMS) was performed from October 2017 to March 2018. Surface observations of SIA at 28 sites were obtained to evaluate the model, which confirmed the biases in the SIA modeling. To identify the source of these biases and reduce them, uncertainty analysis was performed by evaluating the heterogeneous chemical reactions in the model and conducting sensitivity tests on the different reactions. The results suggest that the omission of the $SO_2$ heterogeneous chemical reaction involving anthropogenic aerosols in the model is probably the key reason for the systematic underestimation of sulfate during the winter season. The uptake coefficient of the "renoxification" reaction is a key source of uncertainty in nitrate simulations, and it is likely to be overestimated by the NAQPMS. Consideration of the $SO_2$ heterogeneous reaction involving anthropogenic aerosols and optimization of the uptake coefficient of the "renoxification" reaction in the model suitably reproduced the temporal and spatial variations in sulfate, nitrate and ammonium over North China Plain. The biases in the simulations of sulfate, nitrate, ammonium, and particulate matter smaller than 2.5 μm ($PM_{2.5}$) were reduced by 84.2%, 54.8%, 81.8%, and 80.9%, respectively. The results of this study provide a reference for the reduction in the model bias of SIA and $PM_{2.5}$ and improvement of the simulation of heterogeneous chemical processes.

**Keywords:** simulation evaluation; secondary inorganic aerosol; heterogeneous chemistry

## 1. Introduction

As one of the most densely populated and economically developed regions in China, North China Plain frequently experiences severe haze pollution events with record-breaking

concentrations of particulate matter smaller than 2.5 μm in diameter ($PM_{2.5}$) [1,2], which exert severe adverse impacts on ecosystems and human health [3]. Under the implementation of the Atmospheric Pollution Prevention and Control Action Plan and other control measures, $SO_2$ and $NO_2$ emissions and haze days have been reduced year by year [4]. For example, the haze days in Beijing have decreased from 58 days in 2013 to 15 days in 2018 [5]. However, secondary inorganic aerosol (SIA) still accounts for more than 30% of the $PM_{2.5}$ concentration [6], and the proportion of SIA in $PM_{2.5}$ often increases during haze pollution periods [7,8]. It is commonly accepted that SIA formation contributes to high $PM_{2.5}$ concentrations on hazy days in North China Plain [9–11]. Moreover, SIA has been found to be the leading driving factor for the formation of several fine-particle explosive growth (FPEG) events in Beijing [12]. Therefore, understanding the formation mechanism of SIA and accurately forecasting their variations are crucial for $PM_{2.5}$ pollution control in North China Plain.

Although SIA plays an important role in $PM_{2.5}$ pollution, it remains a great challenge to accurately simulate and predict the SIA concentration due to the complex formation processes of SIA. The SIA concentration depends on a series of complex chemical reactions involving gas, liquid, and heterogeneous phases and various factors, such as the pH [13], aerosol water content [14], relative humidity (RH) [15], solar radiation [16], temperature (T) [15], and precursor concentration [17]. The latest Model Inter-Comparison Study for Asia phase III (MICS-Asia III) conducted a one-year model evaluation of 14 air quality models and found that most of the evaluated models contain large uncertainties in SIA modeling, especially in winter [18]. Other studies have also found a similar phenomenon [8,19,20]. Therefore, this study focuses on the simulation of secondary inorganic aerosols in winter. Furthermore, recent studies have shown that traditional gas- and aqueous-phase chemical reactions hardly explain the high concentration of SIA observed during haze pollution periods [8,21]. Observation studies have indicated that SIA rapidly increase under the conditions of a high RH and a temperature of approximately −5 °C [15]. Heterogeneous chemical processes have been highlighted in several studies as an alternative pathway for SIA formation in winter over North China Plain [21–24]. However, accurate simulation of heterogeneous chemical processes remains controversial and uncertain due to the considerable uncertainty or even the lack of treatment of the uptake coefficient, pH and aerosol water content in models [25]. For example, Cheng et al. [14] combined field measurements and proposed that the missing mechanism of the heterogeneous oxidation of $SO_2$ by $NO_2$ in their model was an important reason for the sulfate underestimation in January 2013 in Beijing. They also pointed out that a high $NH_3$ concentration and high pH value are important prerequisites for this reaction. However, Song et al. [26] revised the steady-state assumption in their model code and found that the pH value simulated by ISORROPIA-II was approximately 4.6. The heterogeneous oxidation of $SO_2$ by $NO_2$ was not important under acidic conditions. These inconsistent conclusions indicate that our current understanding of SIA formation is incomplete.

The uncertainty in SIA simulation has resulted in great uncertainty in the formulation and effectiveness of model-based control measures, and reducing these model uncertainties is urgently needed. Although some work has been performed [8,14,18,23,27,28], most of the verification and evaluation studies were carried out in Beijing at a relatively low spatial resolution, involving scarce data sets and short simulation times. To further evaluate the performance and reduce the uncertainty in SIA modeling, this study employed surface SIA observations at 28 sites in North China Plain and evaluated the SIA modeling performance of the Nested Air Quality Prediction Model System (NAQPMS) with a 15 km horizontal resolution from October 2017 to March 2018. Bias correction tests with heterogeneous reaction schemes and considering their key reaction parameters were carried out to improve the SIA simulation performance. Section 2 introduces the model setting and observation data, Section 3 presents the SIA simulation evaluation results under the different simulation schemes, and Section 4 provides the main conclusions of the paper.

## 2. Experimental Methods

### 2.1. Observation Data

To evaluate the model, daily surface observations of sulfate, nitrate, and ammonium were retrieved from the regional component observation network in North China Plain. The observations were obtained by membrane samplers. All sampling were collected onto mixed cellulose ester filters, which were individually placed in petri dishes and shored at $-20\,^\circ$C immediately after sampling. The filters were weighed with using a micro-electronic balance with a reading precision of 10 μg after 48 h of equilibration inside 25 °C T and 50% RH [29]. There are 38 observation sites in the observation network, but 10 sites were not used in this study due to the very low data availability of these stations. The remaining 28 sites were selected for validation purposes. Furthermore, hourly observations of the $PM_{2.5}$ concentration at 331 sites of the China Environmental Monitoring Centre were employed, and 27 sites the closest to the above SIA sampling sites were selected for analysis purposes. The spatial distribution of the monitoring sites is shown in Figure 1.

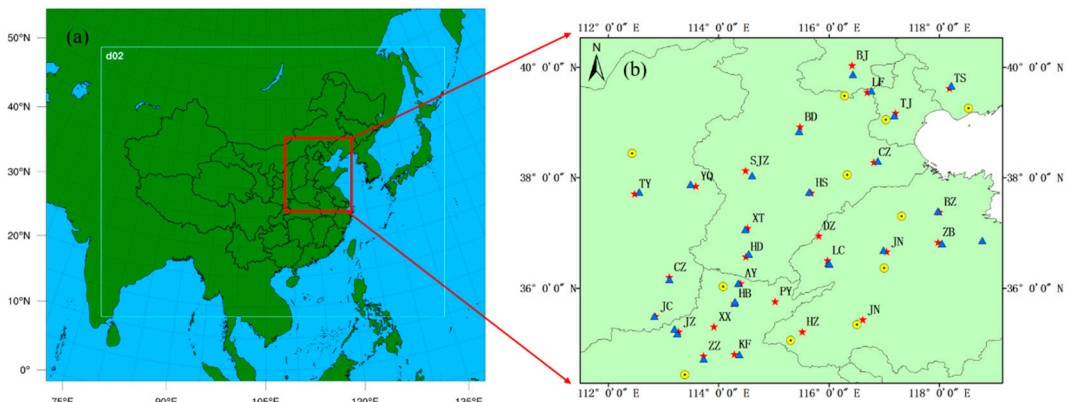

**Figure 1.** Model domain (**a**) and the spatial distribution (**b**) of the monitoring sites used in the model evaluation of the aerosols over North China Plain. The red dots indicate the component observation sites, the yellow dots indicate the $PM_{2.5}$ observation sites, and the blue dots indicate the meteorological observation sites.

### 2.2. Model Setting

The NAQPMS employed in this study was developed by the Institute of Atmospheric Physics, Chinese Academy of Sciences, which has been widely applied in air quality forecasting and numerical simulation research in China [30,31]. The NAQPMS is a three-dimensional Eulerian chemical transport model that includes modules for the simulation of advection and diffusion, dry and wet deposition, emissions, and gaseous, aqueous, and heterogeneous chemical processes of air pollutants. The dry deposition module relied on the Wesely scheme [32]. The wet deposition and aqueous chemistry module adopted the improved chemistry mechanism based on the regional acid deposition model (RADM) [33]. The carbon-bond mechanism version Z (CBM-Z) [34] was applied for the gas-phase chemistry. Partitioning processes and thermodynamic equilibrium determination of sulfate, nitrate, and ammonium between the gas and aerosol phases were performed by the aerosol thermodynamic module ISORROPIA 1.7 [35]. Our study used the NAQPMS to evaluate the SIA in autumn and winter over North China Plain, from 25 September 2017 to 31 March 2018. The first seven days were used as the spin-up time of the NAQPMS. The two nested model domains in this study are shown in Figure 1. The model domains were defined in the Lambert conform projection, and the second domain consisted of 432 × 339 cells with a 15 km horizontal resolution and 20 vertical levels up to 20 km.

### 2.3. Input Data

Hourly meteorological input data of the NAQPMS were provided by the Weather Research and Forecasting (WRF) model version 3.6, which was driven by $1^\circ \times 1^\circ$ reanalysis

data originating from the National Centre for Atmospheric Research/National Centre for Environmental Prediction (NCAR/NCEP). The physics options of the WRF model were configured as follows: the planetary boundary layer was configured with the Yonsei University (YSU) scheme [36], the land surface process selected the Noah land surface process scheme [37], the microphysics scheme relied on the WRF Single Moment 3 (WSM3) simple ice scheme [38], longwave radiation was computed with the rapid radiative transfer model (RRTM) [39], and shortwave radiation was determined with the Dudhia scheme [40]. In the daily meteorological simulations, WRF runs were integrated over individual 36 h periods. Each run included a meteorological spin-up time of the first 12 h, and the data of the remaining 24 h were considered in the NAQPMS. The meteorological simulation results were evaluated against daily observations retrieved from the China Meteorological Data Network (http://data.cma.cn (accessed on 11 September 2020)). The 11 meteorological sites the closest to the component monitoring sites were selected among 166 sites, and the distribution is shown in Figure 1. Figure S1 shows a time series comparison of the observed and simulated temperature, RH, and wind speed at the Tianjin site. In general, the WRF model reproduces the temporal distribution characteristics of the main meteorological factors during the simulation period and provides reliable input data for the NAQPMS.

The adopted emission inventory includes anthropogenic, biomass burning, aircraft, ship, and volcanic emissions. Monthly anthropogenic emissions with a 0.25° spatial resolution were acquired from the Multi-resolution Emission Inventory for China (MEIC) for 2016, and the $NH_3$ emission inventory for China was updated by Peking University. More exhaustive information related to anthropogenic emissions has been reported in Li et al. [41]. Hourly biomass burning emissions were provided by the Global Fire Emission Date base (GFEDv4), and biogenic emissions were calculated with the Model of Emissions of Gases and Aerosols from Nature version 2.04 (MEGAN-v2.04) [42]. Aircraft, ship, and volcanic emissions at a 0.1° spatial resolution were collected from Hemispheric Transport of Air Pollution version 2 (HATP-v2) [43].

### 2.4. Description of the Heterogeneous Chemistry Module

The heterogeneous chemical scheme adopted in this study was based on the settings of Li et al. [44]. To assess the degree of mixing between Asian mineral dust and anthropogenic pollutants during dust storm events, Li et al. [44] considered 28 heterogeneous reactions based on numerous previous works, thereby successfully reproducing the trajectory of long-range dust transport and the chemical evolution of dust particles. The 28 heterogeneous reactions assumed to occur on the surface of aerosols are listed in Table S1 Heterogeneous reactions are commonly parameterized using a pseudo-first-order rate constant, which is calculated with Equation (1), as proposed by Jacob [45]:

$$K_i = \left( \frac{r}{D_i} + \frac{4}{v_i \gamma_i} \right)^{-1} \times A \qquad (1)$$

where $i$ is the reactant for the heterogeneous reaction, $r$ is the effective radius of the particles (m), $D_i$ is the gas-phase molecular diffusion coefficient for reactant $i$ (m$^2$ s$^{-1}$), $v_i$ is the mean molecular speed of reactant $i$, $A$ is the aerosol surface area per unit volume of air (m$^2$ m$^{-3}$), and $\gamma_i$ is the uptake coefficient for reactant $i$, which was set according to Li et al. [44]. At different temperatures, humidity levels, and particle characteristics, the uptake coefficient may vary by several orders of magnitude, and the effect of RH and T on the uptake coefficient was considered for specific particulate matter and gas-phase pollutants [44]. In this paper, the results simulated considering the above 28 heterogeneous chemical reactions were defined as the base run.

## 3. Results and Discussion

### 3.1. Model Evaluation of the Base Run

Figure 2 shows the spatial distributions of the simulated and observed sulfate, nitrate, and ammonium concentrations over North China Plain. In general, the base run underes-



timated the sulfate, nitrate, and ammonium concentrations at most sites throughout the six-month period, especially in heavily polluted areas. The underestimation of the sulfate and ammonium concentrations during the simulated period at 28 sites exceeded 30%, at $-3.7 \, \mu g \, m^{-3}$ and $-3.3 \, \mu g \, m^{-3}$, respectively. The underestimation of the nitrate concentration exceeded 15%, and the model bias (MB) reached $-3.2 \, \mu g \, m^{-3}$. Figure 3 shows the temporal distributions of the model bias of sulfate, nitrate, and ammonium over North China Plain during the simulated period. The polluted period was defined as the period with a daily PM$_{2.5}$ concentration higher than $75 \, \mu g \, m^{-3}$, shown as the red background in Figure 3. As shown in Figure 3, sulfate was generally underestimated, which was relatively severe on the pollution days, especially in December and January. The monthly model biases in December and January were $4.6 \, \mu g \, m^{-3}$ and $6.1 \, \mu g \, m^{-3}$, respectively. Nitrate was also underestimated during the polluted period but was overestimated during the cleaning period. The model bias of nitrate was the largest in March, and the largest daily model bias reached $-29.3 \, \mu g \, m^{-3}$ (84%). Some studies have shown that observed nitrate are usually underestimated due to the sampling artifact caused by the evaporative loss of semi-volatile ammonium nitrate [46,47], but the simulation of nitrate was still significantly underestimated, which further confirms the model biases of nitrate simulation in winter. Similar to sulfate, ammonium mainly exhibited a negative bias. In addition to the large biases in December and January, there was also a notable underestimation in March. The monthly model biases in December, January, and March were $-3.7 \, \mu g \, m^{-3}$, $-4.3 \, \mu g \, m^{-3}$, and $-4.3 \, \mu g \, m^{-3}$, respectively. This could be explained by the joint influence of the sulfate and nitrate simulation processes. The consistent underestimation of sulfate, nitrate, and ammonium suggest that a high uncertainty remained in the SIA simulations.

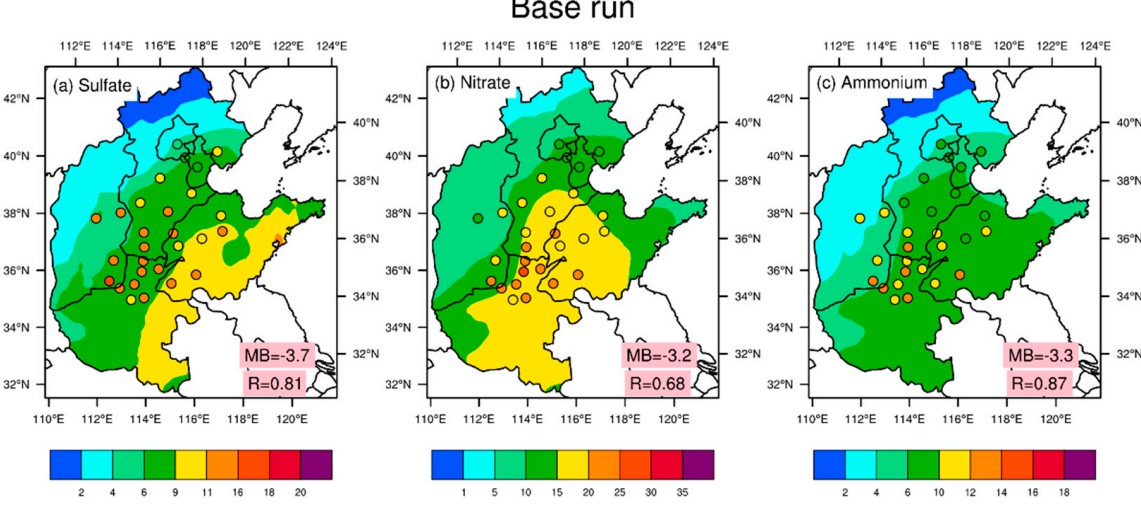

**Figure 2.** Simulated (shaded areas) and observed (solid circles) mean surface sulfate (**a**), nitrate (**b**) and ammonium (**c**) concentrations ($\mu g \, m^{-3}$) at the 28 sites across North China Plain from 1 October 2017 to 31 March 2018. The simulation results are the mean values over six months.

Studies have revealed that high RH and low T levels are advantageous to the occurrence of heterogeneous reactions [15,48]. Combining the time series of the observed T and RH values extracted from the corresponding sites, it was found that the RH level was high and T approached $-5 \, ^\circ C$ in December and January when the model bias in regard to sulfate and ammonium was large. The RH level was also high in March when the model bias in regard to nitrate was the largest. It was found that the base run did not reproduce the SIA peaks during these episodes, probably due to the absence of critical heterogeneous reaction mechanisms that serve as important SIA formation pathways.

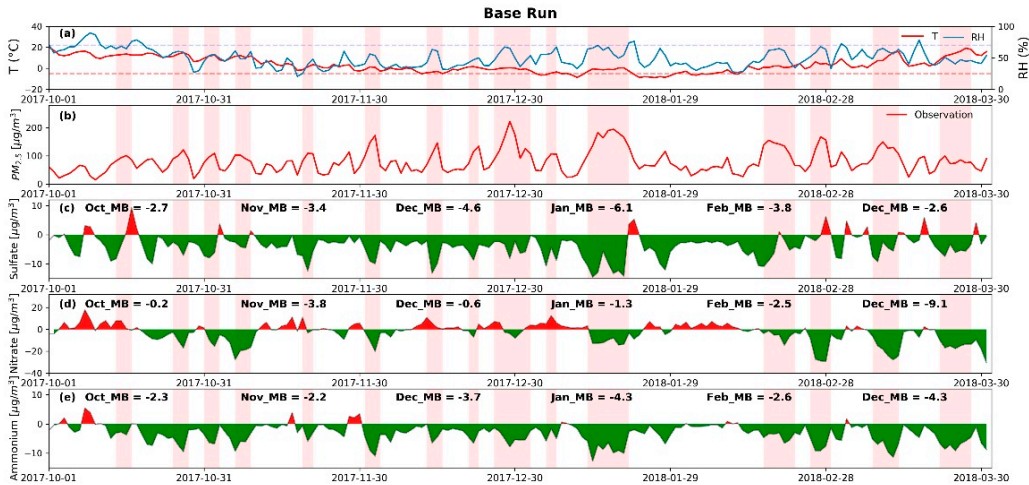

**Figure 3.** (**a**) Observed T and RH at corresponding sites from 1 October 2017 to 31 March 2018. The blue dotted line represents RH equal to 70%, and the red dotted represents the T equal to $-5°$ (**b**) Observed PM$_{2.5}$ at corresponding sites from 1 October 2017 to 31 March 2018. The red background represents the polluted period. (**c**–**e**) Model bias (the mean value of 28 sites) simulated under the base scheme of sulfate, nitrate, and ammonium from 1 October 2017 to 31 March 2018. The red area indicates a positive bias, and the green area indicates a negative bias.

### 3.2. Improvement of the Heterogeneous Chemical Chemistry Module

As discussed in Section 3.1, the SIA simulation results in autumn and winter over North China Plain still exhibited a notable bias, especially the sulfate simulation results, agreeing with our previous evaluation results in the Pearl River Delta region [20]. Our evaluations further confirmed the high uncertainty in the SIA simulations revealed by the model evaluation of the latest MICS-Asia III [18]. Based on the analysis in Section 3.1 and the latest research results [18,23,28], heterogeneous chemical process simulation is likely the key source of the uncertainty in the SIA simulations. In the base run, the heterogeneous chemical process mainly considered the heterogeneous reactions involving sand, sea salt, and black carbon (BC) aerosols and considered the heterogeneous reaction of nitrogen oxides, HO$_2$, HONO, and OH involving sulfate aerosols. However, the heterogeneous reactions critical to sulfate formation have rarely been considered. At present, the model only considers the SO$_2$ heterogeneous reaction involving sand and sea salt. The heterogeneous reaction of SO$_2$ involving anthropogenic aerosols was not considered, which might exert major impacts on sulfate and nitrate formation during winter [8]. Therefore, we performed a sensitivity analysis of a new parameterization scheme of the SO$_2$ heterogeneous reaction involving anthropogenic aerosols according to the base run to evaluate the influence of this reaction on SIA formation.

We defined the SO$_2$ heterogeneous reaction involving anthropogenic aerosols as R29, as expressed in Equation (2). Anthropogenic aerosols are the components in PM$_{2.5}$ except for sand and sea salt. The heterogeneous chemistry was parameterized using the first-order reaction constant proposed by Jacob [45]. We also considered the influence of RH on the uptake coefficient, and $\gamma_{low}$ and $\gamma_{high}$ represented the range of the uptake coefficient of R29, as expressed in Equation (3), where $\gamma_{low} = 2 \times 10^{-5}$ and $\gamma_{high} = 5 \times 10^{-5}$. The uptake coefficient was determined by Zheng et al. [8] through a set of sensitivity experiments, and the value also fell within the estimated range for the SO$_2$ heterogeneous reaction involving particulate matter from $(1.6 \pm 0.7) \times 10^{-5}$ to $(4.5 \pm 1.1) \times 10^{-5}$ [49]. In this paper, the results simulated by the addition of R29 were defined as the run with bias correction scheme I.

$$SO_2(g) + anthropogenic\ aerosols \rightarrow SO_4^{2-} \tag{2}$$

$$\gamma_i = \begin{cases} \gamma_{low} & RH \in [0\%, 50\%] \\ \gamma_{low} + 2\left(\gamma_{high} - \gamma_{low}\right) \times (RH - 0.5) & RH \in [50\%, 100\%] \end{cases} \tag{3}$$

Figure 4 shows time series of the model bias of sulfate, nitrate, and ammonium over North China Plain after the inclusion of R29. The modeled and observed variations of sulfate, nitrate, and ammonium concentrations can be find in Figure S2. The $SO_2$ heterogeneous reaction involving anthropogenic aerosols was found to greatly affect the sulfate concentration, especially in December and January, indicating that R29 improves the simulation of the sulfate concentration by accelerating the $SO_2$ conversion process into sulfate. Ammonium exhibited a similar result with a smaller improvement than that provided by sulfate. However, the addition of R29 aggravated the nitrate underestimation in each month. As $NH_3$ preferentially reacted with $H_2SO_4$ when $H_2SO_4$ and $HNO_3$ both occurred, and the addition of R29 increased the $H_2SO_4$ concentration, leading to a decrease in the $NH_3$ concentration reacting with $HNO_3$ to form nitrate. Overall, the model performance of sulfate over North China Plain is notably improved by including the $SO_2$ heterogeneous reaction involving anthropogenic aerosols in the model. However, the concentration and distribution of nitrate were not captured well, and large underestimations still occurred.

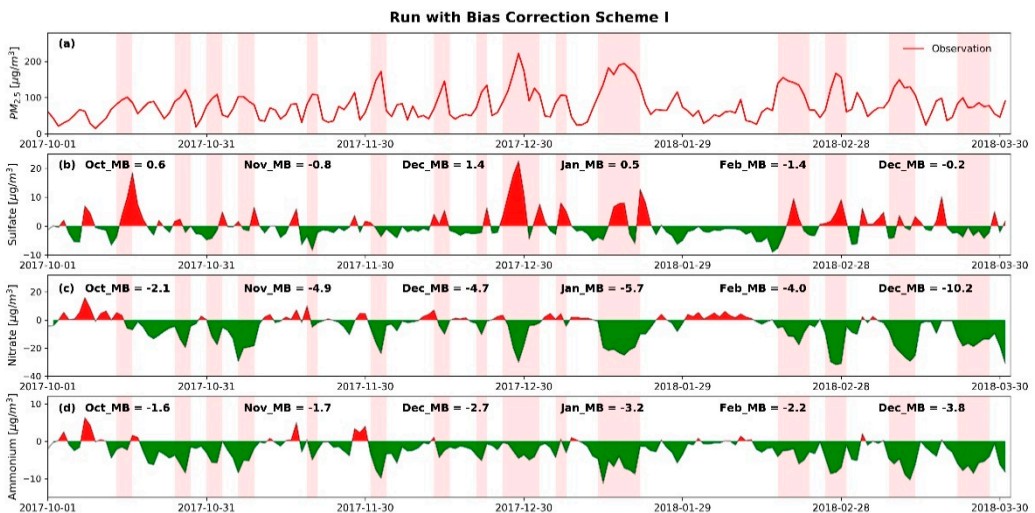

**Figure 4.** (**a**) Observed $PM_{2.5}$ at corresponding sites from 1 October 2017 to 31 March 2018. The red background represents the pollution period. (**b–d**) Model bias (the mean value of 28 sites) simulated under bias correction scheme I of sulfate, nitrate and ammonium from 1 October 2017 to 31 March 2018. The red area indicates a positive bias, and the green area indicates a negative bias.

### 3.3. Sensitivity Analysis and Parameter Optimization of the Heterogeneous Chemistry Module

According to the evaluation of the run with bias correction scheme I in Section 3.2, we found that the updated heterogeneous parameterization scheme of $SO_2$ involving anthropogenic aerosols aggravated the nitrate underestimation phenomenon. To identify the key reactions that affect the nitrate concentration and quantify the relative importance of the different heterogeneous reactions, a set of sensitivity tests was conducted for the 29 heterogeneous reactions considered in our model. The setting of the sensitivity tests is summarized in Table 1. The heterogeneous reactions involving dust, sea salt, BC, and sulfate aerosols were turned off respectively to evaluate the impact of these reactions on the SIA simulations. The simulation period of the sensitivity test was January 2018, when the observed hazy pollution was the most serious. Based on the sensitivity test results listed in Table 1, R2, R8, R9, R29, and the heterogeneous reaction involving sea salt imposed a major effect on the SIA simulations. Among them, R29 contributed the most to sulfate formation, at 45.6%. It was further observed that the heterogeneous transformation of $SO_2$ involving anthropogenic aerosols exerted a much greater impact on sulfate than did the heterogeneous transformation of $SO_2$ involving sand dust and sea salt. The influence of dust and sea salt was not the focus of our study, but further study is recommended. R9 was found to exert a profound impact on nitrate formation, which converts the $HNO_3$ produced

by the heterogeneous reaction involving BC into $NO_2$, as expressed in Equation (4). This reaction is a heterogeneous "renoxification" reaction [50]. The sensitivity test results indicated that turning off R9 resulted in a 77.7% increase in nitrate, indicating that this reaction had a decisive influence on the nitrate simulation results.

$$HNO_3 + BC \rightarrow NO_2 \tag{4}$$

**Table 1.** Effect of the heterogeneous reactions on the SIA simulations.

| | Sulfate | Nitrate | Ammonium |
|---|---|---|---|
| R1 turned off | 0.7% | −1.5% | −0.3% |
| R2 turned off | −1.8% | −16.3% | −8.5% |
| R3 turned off | 0.8% | −0.6% | 0.15% |
| R4 turned off | 0.9% | −0.5% | 0.2% |
| R5 turned off | 1.6% | 3.0% | 2.2% |
| R6 turned off | 0.9% | −0.6% | 0.2% |
| R8 turned off | −12.2% | −20.6% | −16.3% |
| R9 turned off | 8.0% | 77.7% | 38.7% |
| R10 turned off | 0.1% | −2.67% | −1.3% |
| Dust (R11–R22) turned off | 0.8% | −0.6% | 0.2% |
| Sea salt aerosols (SSA) (R23–R28) turned off | −12.6% | 6.7% | −3.5% |
| R29 turned off | −45.6% | 27.9% | −11.8% |

The heterogeneous "renoxification" reaction of $HNO_3$ was proposed by Lary et al. [51] to address the overestimation of $HNO_3$. In recent years, many studies through observation and laboratory simulations have shown that the "renoxification" process is very complicated, which is influenced by various factors, such as reactants, reaction products, ionic characteristics, and atmospheric conditions [52,53], and still contains a high uncertainty [54]. As a result, the "renoxification" reaction mechanism has not yet been well established [50]. Akimoto et al. found that a model considering the "renoxification" reaction of $HNO_3$ with an uptake coefficient of 0.003 yields a more accurate simulation of NO than do other models. However, the uptake coefficient of the "renoxification" reaction in our base scheme was set to 0.02. This value is not only higher than 0.003 but also much higher than the estimated range of the uptake coefficient of heterogeneous chemical reactions in many current models. A high uptake coefficient of the "renoxification" reaction results in more $HNO_3$ being converted into $NO_2$. The overestimated uptake coefficient of the "renoxification" reaction is likely the key source of uncertainty in the notable underestimation of nitrate. Therefore, this paper selected the uptake coefficient of R9 as the key uncertainty factor of the heterogeneous process simulation to conduct a series of sensitivity tests. The parameter settings of the sensitivity test are provided in Table S2. Figure S3 shows the results of the sensitivity test. The uncertainty of the uptake coefficient of R9 imposed little effect on sulfate, and the change in sulfate concentration caused by the uptake coefficient of R9 ranged from 0 to 1.2 $\mu g\ m^{-3}$. The correlation between the simulations and observations remained almost unchanged. However, nitrate was sensitive to the uptake coefficient of R9. With decreasing uptake coefficient from 0.02 to 0, the nitrate simulation changed from underestimation (MB = −5.5 $\mu g\ m^{-3}$) to overestimation (MB = 9.2 $\mu g\ m^{-3}$). When the uptake coefficient of R9 was 0.005, i.e., in case S3, the sulfate, nitrate and ammonium simulation results were highly consistent with the observations, and the nitrate and ammonium underestimations were greatly improved. The value of 0.005 is consistent with 0.003. Moreover, 0.005 is also within the range of the uptake coefficient of the "renoxification" reaction from $1.1 \times 10^{-3}$ to $2 \times 10^{-1}$ estimated in other studies [55,56].

We defined S3 as the run with bias correction scheme II, namely, consideration of R29 and an R9 uptake coefficient of 0.005. Figure 5 illustrates the ability of bias correction scheme II to capture the spatial distribution of the observed SIA in autumn and winter over North China Plain. The model results obtained by the addition of the $SO_2$ heterogeneous

reaction involving anthropogenic aerosols and correction of the uptake coefficient of R9 indicate a greatly improved model performance of sulfate, nitrate and ammonium, especially in heavily polluted areas. The model biases of sulfate, nitrate and ammonium decreased by 84.2%, 54.8%, and 81.8%, respectively. The correlation coefficients of the sulfate, nitrate and ammonium modeling results also increased. Overall, the run with bias correction scheme II shows a similar magnitude and spatial distribution characteristics as those of the observations.

**Figure 5.** Same as Figure 2 but the results are simulated under bias correction scheme.

Figure 6 compares the temporal variations in the simulated and observed sulfate, nitrate, ammonium and $PM_{2.5}$ concentrations in North China Plain in the base run and the run with bias correction scheme II. The sulfate concentration simulated under the two different schemes shows similar patterns during the clean periods but major differences during the polluted periods, especially in winter when the reaction rates of gas- and aqueous-phase chemical processes are very low [7,8]. The sulfate concentration simulated under bias correction scheme II increased by 10–23 $\mu g\ m^{-3}$ (49–78%) from 13 to 20 January, which explained the observed rapid growth in the sulfate concentration during this period. Similar phenomena were also found for the other sulfate and ammonium peaks, indicating that the uncertainty in heterogeneous chemistry is an important source of the underestimation of sulfate and ammonium. It should be noted that the nitrate concentration simulated under the base scheme was greatly underestimated in March, and the addition of R29 aggravated this underestimation. The modification of the uptake coefficient of R9 greatly improved the nitrate simulation results, and the increase in the nitrate concentration ranged from 8 to 14 $\mu g\ m^{-3}$ (20–62%) from 10–14 to 22–26 March. These results indicate that the observed nitrate underestimation was probably associated with the overestimation of the uptake coefficient of the "renoxification" reaction. The nitrate simulation was improved on the whole. However, the NMB of nitrate simulated by the bias correction scheme II was larger than the base run. The nitrate was overestimated in early October after correction of the uptake coefficient of R9, and NMB has numerical asymmetry when the observation is too large or too small [4], which leads to large NMB. It revealed that the uptake coefficient of R9 setting as 0.005 may not be appropriate for all situations. It suggests that the uptake coefficient of R9 can be set at different values under different conditions, which needs further study, and 0.005 is a recommendation during heavy pollution. Overall, bias correction scheme II greatly improved the accuracy of the SIA simulations and ensured that the model results were more consistent with the observed peaks. Moreover, the reasonable simulation of SIA by chemical transport models also considerably improved the $PM_{2.5}$ simulations, of which the model bias was reduced by 80.9% and the root mean square error (RMSE) was reduced by 15.5%, thus further verifying the rationality of bias correction scheme II.

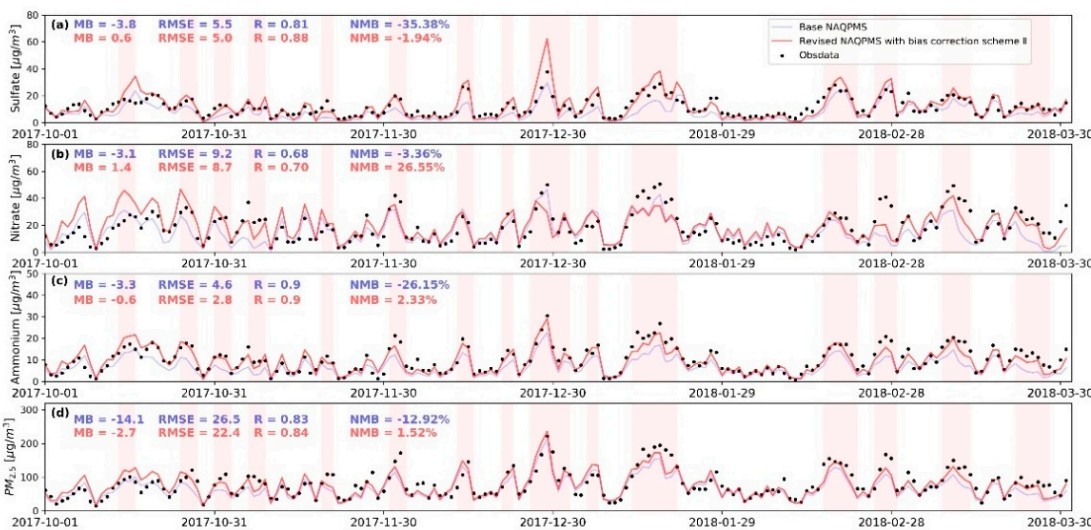

**Figure 6.** Comparison of the simulated and observed (black solid cycles) daily sulfate (**a**), nitrate (**b**), ammonium (**c**), and PM$_{2.5}$, (**d**) concentrations averaged over the 28 monitoring sites in North China Plain from 1 October 2017 to 31 March 2018. The red background represents the polluted period. The black dots correspond to the observations, and the red and blue lines indicate the simulations of the base run and the run with bias correction scheme II, respectively.

## 4. Conclusions

In this study, we performed a six-month simulation over North China Plain at a 15 km horizontal resolution and compared the results to surface observations acquired at 28 sites. The results showed that the model notably underestimated sulfate, nitrate and ammonium, especially in winter with a high RH and low T. Through a sensitivity analysis of various heterogeneous processes, a new parameterization scheme considering the SO$_2$ heterogeneous reaction involving anthropogenic aerosols and optimization of the uptake coefficient of the "renoxification" reaction was incorporated into the model, which yielded a reasonably good agreement in terms of the magnitude and variation in the SIA and PM$_{2.5}$ concentrations in autumn and winter over North China Plain. Our study revealed that the SO$_2$ heterogeneous reaction involving anthropogenic aerosols might be the key pathway of sulfate formation in winter in North China Plain, and the omission of this pathway in model could lead to a notable underestimation of sulfate in winter. In addition, the uptake coefficient of the heterogeneous reaction was found to impose an important influence on the nitrate simulations. However, many of the uptake coefficients currently used in the model are based on empirical estimates retrieved from relevant studies. Measurement data of the uptake coefficient under actual atmospheric conditions are still very scarce, which is an important source of the uncertainty in SIA simulations at present. Therefore, further investigation of the uptake coefficient must be carried out in the future.

Although the bias correction scheme adopted in this paper greatly reduces the model bias of SIA in autumn and winter over North China Plain, certain limitations remain in this paper. First, there is still no mature heterogeneous simulation scheme similar to gas-phase chemistry modules (such as CBM-Z) since SIA formation in winter remains very controversial at present. The contribution of our work is to provide a reference for the improvement of heterogeneous chemical process simulations, but more validation tests are still needed to evaluate the applicability of this scheme in other regions and seasons. In addition, certain critical formation pathways of SIA in different regions are still lacking in the current model [24,57]. For example, SO$_2$ oxidation catalyzed on BC aerosols in the presence of NO$_2$ and NH$_3$ [5]. Incorporation of a more comprehensive SIA simulation scheme into models and systematic assessment of its applicability under different environmental conditions are urgent tasks.

**Supplementary Materials:** The following are available online at https://www.mdpi.com/article/10.3390/atmos12050578/s1, Figure S1: Time series of the simulated (red) and observed (black) daily averaged meteorological parameters at the Tianjin site from 1 October 2017 to 31 March 2018; Figure S2: Comparison of the simulated and observed (black solid cycles) daily sulfate (a), nitrate (b), ammonium (c), and PM$_{2.5}$ (d) concentrations averaged over the 28 monitoring sites in North China Plain from 1 October 2017 to 31 March 2018. The black dots correspond to the observations, and the red and blue lines indicate the simulations of the base run and the run with bias correction scheme I, respectively; Figure S3: Rose diagram of the statistical indicators for the simulation of sulfate, nitrate and ammonium in the different cases of R9. The radius of each sector represents the degree of correlation between the simulations and observations, the radian indicates the RMSE, the different colors indicate the MB, and the value in the shaded block is the MB value; Table S1: The 28 heterogeneous reactions considered in the NAQPMS; Table S2: Sensitivity tests of the uptake coefficients of R9; Table S2: Sensitivity tests of the uptake coefficients of R9.

**Author Contributions:** X.T.: conducted the design of this study. X.D., W.W., Q.W. (Qian Wang) and D.C.: provided the observation data. Q.W. (Qian Wu) and L.K.: provided modeling data. Q.W. (Qian Wu): analyzed and interpreted the data; drafted the article and revised it. X.T., L.K. and M.L.: revised the article critically. Z.L., L.W., X.P., J.L., J.Z. and Z.W.: reviewed and commented on the paper. All authors have read and agreed to the published version of the manuscript.

**Funding:** This work was supported by the National Key R&D Program (Grant Nos. 2018YFC0213503 & 2017YFC0212603), the National Natural Science Foundation (Grant No. 41875164), Guangdong Provincial Science and Technology Development Special Fund (Grant No. 2017B020216007), and the CAS Information Technology Program (Grant No. XXH13506-302).

**Institutional Review Board Statement:** Not applicable.

**Informed Consent Statement:** Not applicable.

**Data Availability Statement:** Not applicable.

**Conflicts of Interest:** The authors declare that they have no conflict of interest.

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
