# Peer review of "Evaluation and Bias Correction of the Secondary Inorganic Aerosol Modeling over North China Plain in Autumn and Winter"

_atmosphere, doi:10.3390/atmos12050578_

Round 1

Reviewer 1 Report

The article” Evaluation and Bias Correction of the Secondary Inorganic Aerosol Modelling Over North China Plain in Autumn and Winter” highlights the idea that the omission of the SO2 heterogeneous chemical reaction involving anthropogenic aerosols in the model is probably the key reason for the systematic underestimation of sulphate during the cold season.

The Secondary inorganic aerosol study was done between October 2017 and March 2018, in the North China Plain with the 15-km resolution using Nested Air Quality Prediction Model System.

The main contribution of this paper is the fact that the uptake coefficient of the “renoxification” reaction was incorporated into the model and parameterization scheme consider the SO2 heterogeneous reaction.

The sources of data are multiple, such as: the meteorological simulation results were evaluated against daily observations retrieved from the China Meteorological Data Network, hourly meteorological input data of the NAQPMS were provided by the Weather Research and Forecasting, data originating from the National Centre for Atmospheric Research/National Centre for  Environmental Prediction, Monthly anthropogenic emissions with a 0.25° spatial resolution were acquired from the Multi-resolution Emission Inventory for China, the NH3 emission inventory for China was updated by Peking University, Hourly biomass burning emissions were provided by the Global Fire Emission Date base, etc.

The simulation model was performed at a 15-km horizontal resolution and it was compared with the results of surface observations acquired at 28 sites. 

If we emphasize the weaknesses of the paper, then it should be mentioned that the analysis period is only 6 months. But this period can be extended, and the method can be applied to other pollutants.

As mentioned by the authors of this article, a number of other simulation schemes can be added into models under different environmental conditions.

Row 208 ”(the mfean value of 28 sites)„ - should be ”mean”

Reviewer 2 Report

Please consider the revisions suggested in the attached file.

Reviewer 3 Report

The paper “Evaluation and Bias Correction of the Secondary Inorganic Aerosol Modeling over North China Plain in Autumn and Winter” (Manuscript ID atmosphere-1199624) is original and interesting, relevant to the journal, presenting the results obtained by the research team for a six-month simulation over North China Plain at a 15-km horizontal resolution NAQPMS and compared the results with surface observations of secondary inorganic aerosol (SIA) acquired at 28 sites.

The prediction of SIA concentration variations and understanding the complex mechanism of aerosol formation are very important issues for PM2.5 pollution control. The paper aimed to improve the SIA simulation performance by adopting bias correction tests with heterogeneous reaction schemes and considering key reaction parameters in order to reduce the biases in SIA modeling in autumn and winter in the target region.

 Some small mistakes in the references list should be corrected before acceptance:

1). The title of the journal is not specified at the references nos. 1,10,12,20,24,28,34,43,44,45,47 and 49;

2) At some references the name of the last author is not correct and before the cited paper title an abbreviation is written (see, for example: T.e.o.t.T.E. at [12]; J.J.o.G.R.A. at [43],[47]; J.J.o.A.C. at [44]; J.G.R.L. at [45], etc.);

3) At some references ([5],[17]) it appears in text “%J”. 

Thus, I recommend its publication in the journal “Atmosphere” after minor revision.
